# Improved Resource State for Verifiable Blind Quantum Computation

**DOI:** 10.3390/e22090996

**Published:** 2020-09-07

**Authors:** Qingshan Xu, Xiaoqing Tan, Rui Huang

**Affiliations:** College of Information Science and Technology, Jinan University, Guangzhou 510632, China; xuqingshan1008@stu2018.jnu.edu.cn (Q.X.); hrui01@stu2019.jnu.edu.cn (R.H.)

**Keywords:** blind quantum computation, quantum verification, delegated quantum computation

## Abstract

Recent advances in theoretical and experimental quantum computing raise the problem of verifying the outcome of these quantum computations. The recent verification protocols using blind quantum computing are fruitful for addressing this problem. Unfortunately, all known schemes have relatively high overhead. Here we present a novel construction for the resource state of verifiable blind quantum computation. This approach achieves a better verifiability of 0.866 in the case of classical output. In addition, the number of required qubits is 2N+4cN, where *N* and *c* are the number of vertices and the maximal degree in the original computation graph, respectively. In other words, our overhead is less linear in the size of the computational scale. Finally, we utilize the method of repetition and fault-tolerant code to optimise the verifiability.

## 1. Introduction

Scalable quantum computing still has a long way to go, while quantum computing in cloud mode is relatively reasonable. The scenario is that a client who only has access to classical computation and a limited quantum device used for preparing or measuring single qubits delegates a computation task to an untrusted server with a full-fledged quantum computer. In addition, the client’s input, output, and computation remain private to the server. Such secure quantum computing protocols are called blind quantum computing (BQC) [1,2,3,4,5,6,7,8,9,10,11,12,13,14,15,16]. However, how can a client verify the outcome of the computation sent by a server when a quantum experiment solves a problem which is proven to be intractable for classical computers? Fortunately, there has been a lot of progress in the development of verification protocols [17,18,19,20,21,22,23,24,25,26,27,28,29]. The goal of verifiable universal blind quantum computation (VUBQC) is to detect deviation with high probability when the server behave dishonestly and reject his output. Here, the VUBQC scheme we consider is based on constructing the delegated computation to include certain traps in such a way that the computation is not affected, while revealing no information to the device [17]. Then one can verify that the computation has been performed correctly, with exponentially small probability of error.

There are two important properties in the verification protocols [28]. The first one is verifiability, which means the maximal probability for the output of the protocol to be incorrect and the client accepting. The other one is correctness, which means the minimal probability for the client obtaining the correct outcome when the server behaves honestly. Especially, we characterize the client as a verifier and the server as a prover. The term “verifiable” for VUBQC is related to notions of completeness and soundness in the context of interactive-proof system. Given a problem that is classically intractable, the verifier can accept a correct solution with high probability and reject a invalid solution with high probability at the end of the interaction with the prover. Note that even if the verifier accept the outcome sent by the prover, the outcome may be still incorrect. However, the probability that the verifier accepting a wrong outcome can be reduced to a value approaching 0 through some improvements for VUBQC.

In reality, exploring a verification protocol with arbitrarily small verifiability while keeping the cost of resource optimal is still an opening problem. Some progress has been made in this regard. In [17], dotted-complete graph was used for resource construction in verification protocol. It can achieve verifiability $\epsilon ={\left(5/6\right)}^{\left\lceil 2d/5\right\rceil }$, where *d* is the distance of error correcting code used in the protocol. However, the overhead of verification protocol is quadratic in the size of the computation. In [25], a optimised resource construction using dotted-triple graph was proposed, where the number of traps can be a constant fraction of the total number of qubits. It can obtain verifiability $\epsilon ={\left(8/9\right)}^{\left\lceil d/18\right\rceil }$. More importantly, it only requires a linear overhead in the size of the computation.

The verification scheme we present here makes use of similar elements as suggested in [17], trap computations are used to detect errors and a fault-tolerant encoding of the computation is used to amplify the detection rate. Compared with [17], we construct a sandglass-like resource state such that the overhead is linearly related to the size of the computation. In addition, compared with [25], not only do we just need fewer qubits, but also achieve a better verifiability.

The remainder of the paper is organized as follows. In Section 2 we give some basic notions about verifiable universal blind quantum computation. Next, in Section 3 we give the process of our sandglass-like resource state construction. Then in Section 4 we propose a verifiable blind quantum computation protocol with sandglass-like resource state and analyse the correctness and verifiability of the protocol. For classical output case and quantum output case, in Section 5 we propose an improved scheme to improve verifiability. We finally conclude, in Section 6, with some discussions and open problems.

## 2. Preliminaries

We briefly present the relevant concepts used in describing VUBQC protocols. The first one is the model of measurement-based quantum computation (MBQC) [30,31,32]. Different from the traditional quantum circuit model, in MBQC a given computation is performed by measuring qubits from a large entangled state. This special entangled state consists of qubits prepared in the state $|+\rangle =(|0\rangle +|1\rangle )/\sqrt{2}$, entangled using CZ =$I⊗|0\rangle \langle 0|+Z⊗|1\rangle \langle 1|$ operations. The entangled state is also known as graph state, which can be determined by a given graph. In other words, given an undirected graph *G* with *n* vertices i∈V and edges $\left(i,j\right)\in E$, the graph state $|G\rangle$ that corresponds to *G* is defined by $|G\rangle =\left(\prod_{\left(i,j\right)\in E}CZ_{ij}\right){|+\rangle }^{⊗n}$, where $CZ_{ij}$ is the CZ operation acting on vertices sharing the edge $\left(i,j\right)$. Then they are measured in the basis $\left\{|{+}_{\phi }\rangle =\left(|0\rangle +e^{i\phi }|1\rangle \right)/\sqrt{2},\;|{-}_{\phi }\rangle =\left(|0\rangle -e^{i\phi }|1\rangle \right)/\sqrt{2}\right\}$, where the measurement angle is $\phi \in \left\{0,\pi /4,\cdots ,7\pi /4\right\}$ depending on outcomes of previous measurements.

The second part is blind quantum computing [2], which is based on the MBQC model. The protocol runs as follows: (1) Randomly rotated single-qubit states ${\left\{|{+}_{\theta_{j}}\rangle =\left(|0\rangle +e^{i\theta_{j}}|1\rangle \right)/\sqrt{2}\right\}}_{j=1}^{N}$ are prepared by Alice, where $\theta_{j}\in \left\{0,\pi /4,\cdots ,7\pi /4\right\}$ is a random angle, and then Alice sends them to Bob. (2) Bob creates a certain graph state called the brickwork state [2] by entangling obtained states with CZ operations. (3) Alice calculates the measurement angle depending on outcomes of previous measurements and sends it to Bob. (4) Bob performs the measurement in the angle sent by Alice, and returns the measurement result to Alice. (5) Alice and Bob repeat (3) and (4) until all qubits of the brickwork state are measured. If Bob behaves honestly, Alice obtains the correct outcome of desired quantum computation. Furthermore, whatever malicious Bob does, Bob learns nothing about computation’s input, output, and algorithm.

The last one is the VUBQC protocol [17], which augments BQC with the ability to detect malicious behaviour of server (Bob). Because no entanglement is created when CZ operation acts on state |0〉 or |1〉. One can randomly choose a $|{+}_{\theta }\rangle$ qubit (trap qubit) whose neighbours are computational basis states (dummy qubits) such that this qubit is disentangled from the rest qubits in the graph state. Then measuring this trap qubit in θ angle will obtain the deterministic outcome. In [17], the cylinder brickwork state was used such that there are only disentangled trap qubits in a product state with a brickwork state left after entanglement operations are applied by the Bob. Due to the the positions of the traps and dummies are unknown to Bob, the blindness of the protocol is guaranteed. The verifier (Alice) uses trap qubits as traps to test that the prover (Bob) performs desired quantum operations.

## 3. Sandglass-Like Resource State Construction

We now proceed to construct sandglass-like resource state in a manner similar to the construction of the dotted-triple graph state of [25]. As mentioned in MBQC, given a graph *G* we can obtain a corresponding graph state |G〉, which is used to perform a universal quantum computation. We call *G* a base graph. Then we use the base graph *G* to construct a sandglass-like graph S(G) whose corresponding graph state |S(G)〉 achieves verifiable quantum computation. Furthermore, some operations will be performed on a coloured version of the sandglass-like graph S(G) in order to obtain a subgraph used for computation and a subgraph used for traps. Because the selection of computation subgraph and trap subgraph is unknown to the prover, security of the scheme is protected. The same as [25], our construction of subgraphs is local. However, our method requires less qubits and obtains a better verifiability.

We then give specific definitions and related properties of the sandglass-like resource state. Since the construction of resource state depends completely on the sandglass-like graph (each vertex represents a qubit and each edge represents a CZ entanglement operation), we only need to consider the construction of the sandglass-like graph.

According to reference [17], the dotting operator on graph *G* is defined to be the operator that transforms a graph *G* to a new graph denoted by D(G) by replacing every edge in *G* with a new vertex connected to the two vertices originally joined by that edge. Given an arbitrary base graph *G*, the construction procedure of the sandglass-like graph S(G) is described as follows.

(1)A base graph *G* consists of vertices v∈V(G) and edges e∈E(G).(2)For each vertex $v_{i}\in V\left(G\right)$, we define a set of two new vertices $P_{v_{i}}=\left\{p_{1}^{v_{i}},p_{2}^{v_{i}}\right\}$, where $p_{k}^{v_{i}}$ represents the *k*th vertex of the set $P_{v_{i}}$.(3)For each edge $e_{ij}\in E\left(G\right)$, which connects the vertices $v_{i}$ and $v_{j}$, we define a set of four edges $E_{ij}$ that connect vertices in the set $P_{v_{i}}$ with the vertices in the set $P_{v_{j}}$. More concretely, the set $E_{ij}$ consists of four edges $e_{ij}^{11}$, $e_{ij}^{12}$, $e_{ij}^{21}$, and $e_{ij}^{22}$, where $e_{ij}^{mn}$ represents an edge connecting the vertex $p_{m}^{v_{i}}$ and the vertex $p_{n}^{v_{j}}$.(4)We define an intermediate graph I(G) to be a graph consisting of vertices ${⋃}_{v_{i}\in V\left(G\right)}P_{v_{i}}$ and edges ${⋃}_{e_{ij}\in E\left(G\right)}E_{ij}$ described in steps (2) and (3). We perform the dotting operator on the intermediate graph I(G) resulting in a sandglass-like graph denoted by S(G).

Note that the sandglass-like graph S(G) is actually equal to D(I(G)). An example of the construction of sandglass-like graph S(G) is illustrated in Figure 1. The base graph *G* considered in this example consists of four vertices and three edges, as shown in the Figure 1a. The corresponding intermediate graph I(G) is shown in the Figure 1b. Furthermore, Figure 1c gives the sandglass-like graph S(G) corresponding to the base graph G. According to the construction method, total number of vertices in the sandglass-like graph is $\left|V(S(G))\right|=2\left|V(G)\right|+4\left|E(G)\right|$. We therefore only need (2N+4cN) qubits for our verifiable quantum computation, where *N* is the number of qubits for universal quantum computation and *c* is the maximum degree of the base graph. Our construction can apply to other graph states. Since the basic unit of any graph state is two qubits entangled by a CZ gate, the construction procedure of our sandglass-like graph is exactly aimed at a transformation to each basic unit.

Once we have the sandglass-like graph S(G) we can color it for subsequent break operations and bridge operations. We call the set of vertices $P_{v_{i}}$ a primary set. In addition, we say that the vertices in each primary set are primary vertices. Similarly, we denote the set of four vertices related to each edge $e_{ij}$ as an added set $A_{e_{v_{i},v_{j}}}$, and say that the vertices in each added set are added vertices. Similar to the trap-coloring in [25], our definition about trap-coloring of the sandglass-like graph S(G) satisfies the following conditions.

(1)Primary vertices are coloured in one of the three colours of white, red or green.(2)Added vertices are coloured in one of the three colours of white, red or green.(3)One of vertices in each primary set $P_{v_{i}}$ is uniformly at random chosen to be colored in green. The remaining one vertex of $P_{v_{i}}$ has probability α to be colored in red and probability 1−α to be colored in white, where α is an appropriate constant and 0<α<1.(4)The colours of the primary vertices determine the colours of the added vertices. These added vertices connecting primary vertices of different colours are white. These added vertices connecting both green primary vertices are green. Moreover, these added vertices connecting both white primary vertices are red.

Since the color of the added vertices depends on the color of the primary vertices, one may have no red vertex in each primary set $P_{v_{i}}$ or added set $A_{e_{v_{i},v_{j}}}$. A specific example of trap-coloring is given in Figure 2a.

While the construction and the coloring principle of the sandglass-like graph is public, the specific coloring scheme is completely decided by Alice (the client) so that Bob (the server) can not know which vertex is green or red or white. Every vertex has the possibility to be coloured in red (trap qubit). In addition, the coloring of every primary set is independent from the coloring of other primary sets, and the coloring of every added set depends on the coloring of two adjacent primary sets. These features make the security proof of [25] still applicable for our analysis.

Our inspiration comes from that we keep the computation qubits (green vertices) hidden to an untrusted client while increasing the probability that the qubit (vertex) that any attack acts on is a trap qubit (red vertex) such that any attack has a higher probability to be detected. Specifically, compared with [25] whose such a detection probability is 1/3 for each primary set and 1/9 for each added set, our detection probability is α/4 for each primary set and ${\left(1-\alpha \right)}^{2}/4$ for each added set. The lower detection probability obtained when α is $2-\sqrt{3}$ crucially leads to our better verifiability for the case of classical output (see Theorem 2 in Section 4). Note that we actually trade certain symmetry (an arbitrary qubit is uniformly and randomly coloured in any one of the three colours of white, red or green) to obtain less resource overhead. However, as we will see later, this asymmetry just cause slightly inferior verifiability for the case of quantum output.

In what follows we show how to get computation subgraph and trap subgraph from the colored sandglass-like graph. To do this, we need to introduce the break and bridge operations in [17].

The bridge operator on a vertex *v* of degree 2 on graph *G* is defined to be the operator which connects the two neighbors of *v* and then removes vertex *v* and both adjacent edges from *G*. The break operator on a vertex *v* of graph *G* is defined to be the operator that removes vertex *v* and all adjacent edges from *G*.

As shown in Figure 3, the break and bridge operators are demonstrated, respectively. In Figure 3a the break operator acting on vertex $v_{2}$ removes vertex $v_{2}$ and edges $e_{12}$,$e_{23}$ resulting in isolated two vertices $v_{1}$,$v_{3}$. In Figure 3b the bridge operator acting on vertex $v_{2}$ connects vertices $v_{1}$,$v_{3}$ with a new edge $e_{13}$ and removes vertex $v_{2}$ and edges $e_{12}$,$e_{23}$ resulting in two direct connected vertices.

Note that both break and bridge operations on a graph have corresponding implementations of quantum form [17]. To clarify this, if we measure any qubit in a graph state in Pauli *Z* basis, we will get a state obtained from the graph, in which the measured vertex and its adjacent edges are removed, up to local Pauli *Z* corrections. It is equivalent to the break operation. However, what we use more frequently is another equivalent method. In other words, we set the qubit that the break operator acts on to be a dummy qubit, where the dummy qubit is in the state |0〉 or |1〉. Depending on the specific value of the dummy qubit, a Pauli *Z* rotation on all the neighboring qubits in the graph will be introduced after the entanglement operation is performed. As for the bridge operation, if we measure any qubit in Pauli *Y* basis, we will obtain the graph state corresponding to the graph, in which the measured vertex and its adjacent edges are removed and a new edge connecting the adjacent vertices is created, up to local *Z* rotations by π/2 or −π/2.

Now we move on the generation process of the computation subgraph and trap subgraph. Given a colored sandglass-like graph S(G), we perform break operations on the white vertices and bridge operations on the green added vertices (green square vertices) such that we can obtain a computation subgraph and a trap subgraph, as illustrated by Figure 2b. Further more, the red vertices and green vertices are actually trap qubits and computation qubits, respectively. Note that in Figure 2b we preserve the green square vertices for matching computation qubits with trap qubits (dashed circle).

It is noteworthy that our sandglass-like graph draws the same conclusion as Theorem 1 in [25], which will be used in Section 5. To interpret this, we introduce relevant concepts in [25]. We define the base-location of a vertex *f* of the sandglass-like graph S(G) to be the set $P_{v}$ or $A_{e}$ that contains *f* in S(G). Given a sandglass-like graph S(G) and a collection of *n* base-locations E, we call the set E independently colourable locations (ICL) if the choice of colours within any set corresponding to a base-location in E is independent from the choice of colours in sets corresponding to other base-locations in E.

**Lemma** **1.**
*Given a set S consisting of n base-locations in the sandglass-like graph S(G) and assume that the base graph G has maximum degree c. Then there exist a subset $S^{\prime }\subseteq S$ such that $S^{\prime }$ is independently colourable locations and contains at least $\left|S^{\prime }\right|=\frac{n}{2c+1}$ base-locations.*


**Proof.** From the graph *S* with *n* locations, an ICL subset $S^{\prime }$ can be found as follows. From our construction of the sandglass-like graph, a local-colouring of an added base location corresponds to a local-colouring of both adjacent primary base-locations. Then the necessary and sufficient condition of ICL is obtained. In other words, a set of *n* base-locations E is ICL if and only if for all pairs i,j∈E the sets $\epsilon_{i}\cap \epsilon_{j}=⌀$ (Lemma 3 of [25]), where $\epsilon_{i}=\left\{i\right\}$ if the base-location *i* is primary and $\epsilon_{i}=N_{S\left(G\right)}\left(i\right)$ if the base-location *i* is added.One the one hand, if $S^{\prime }$ contains a primary base-location $v_{i}$, then all its adjacent added base-locations (the maximal number is *c*) $a_{ij}$,$a_{ik}$ will be excluded, as shown in Figure 4a. On the other hand, if $S^{\prime }$ contains an added base-location $a_{ij}$, then all its adjacent primary base-locations $v_{i}$,$v_{j}$ and the adjacent added base-locations $a_{ik}$,$a_{im}$,$a_{jn}$ of the primary base-locations $v_{i}$,$v_{j}$ will be excluded, as shown in Figure 4b. The number of excluded base-locations is at most 2c.As a result, in the worst case there exists an ICL subset $S^{\prime }$ with at least $\frac{n}{2c+1}$ base-locations. □

## 4. Verifiable Blind Quantum Computation with the Sandglass-Like Resource State

In this section, similar to [17,25], we present our verifiable blind quantum computation protocol. However, we use our sandglass-like graph state as the resource state of verifiable blind quantum computation. In addition, compared with [17,25], the verifiability and overhead of our protocol are optimised.

The essential idea of verification protocol is that the trap-colouring chosen by Alice (verifier) is unknown to Bob (prover) so that malicious Bob is difficult to deviate the computation while keeping the trap qubit untouched.

Recall the main procedures for VUBQC [17]. Alice converts a computation task to a graph $G^{\prime }$, where corresponding graph state $|G^{\prime }\rangle$ consists of computation qubits, dummy qubits and trap qubits. In addition, each qubit of $|G^{\prime }\rangle$ has a measurement angle $\phi_{i}$ called computation angle, where $\phi_{i}\in A=\left\{0,\pi /4,\cdots ,7\pi /4\right\}$ for all computation qubits and dummy qubits and $\phi_{i}=0$ for all trap qubits. Alice then prepares states $|{+}_{\theta_{i}}\rangle$ for all computation qubits and trap qubits and computational basis states |0〉 and |1〉 for all dummy qubits. Alice sends all these qubits to Bob, who then entangles them to obtain the graph state $|G^{\prime }\rangle$. Alice sends the practical measurement angle $\delta_{i}={\phi^{\prime }}_{i}+\theta_{i}+r_{i}$ in the measurement order to Bob, where ${\phi^{\prime }}_{i}$ is the updated computation angle depending on $\phi_{i}$ and outcomes of Bob’s previous measurements s, $\theta_{i}\in A$ is used to encrypt measurement angle $\phi_{i}$ and $r_{i}\in \left\{0,1\right\}$ is used to encrypt the outcome of measurement. Especially, $\delta_{i}=\theta_{i}+r_{i}$ for all trap qubits and $\delta_{i}\in A$ for all dummy qubits. When Bob’s each trap measurement outcome $b_{t}$ is equal to expected value $r_{t}$, the outcomes of measurements are accepted and corrected by Alice to obtain real results of the computation task.

Here, in our scheme the graph $G^{\prime }$ is replaced by sandglass-like graph S(G), $\phi_{i}$ is equal to π/2 for all the added green vertices required to be measured in Pauli *Y* basis for performing the bridge operators and the function of calculation $\delta_{i}$ is set as $C\left(i,\phi_{i},\theta_{i},r_{i},x_{i},s\right)$. We therefore give our verification protocol as shown in Protocol 1.
**Protocol 1** Verifiable blind quantum computation with sandglass-like resource state.**Alice’s resources:**(1) A graph *G* with *N* vertices for performing the desired computation task in MBQC mode. The coloured sandglass-like graph S(G) with at most 2N+4cN vertices, where *c* is the maximal degree of the base graph *G* and labeling of vertices is known to Alice and Bob.(2) The positions of dummy qubits for the break operations, set *D*, chosen to be positions of all white vertices. The positions of trap qubits chosen to be all red vertices. The positions of computation qubits chosen to be all green vertices, where green square vertices are used to perform bridge operations.(3) An *l*-qubit input state |I〉.(4) A sequence of measurement angles ϕ=${\left(\phi_{i}\right)}_{1\leq i\leq \left(4N+6cN\right)}$ with $\phi_{i}\in A=\left\{0,\pi /4,\cdots ,7\pi /4\right\}$. 2N+4cN random variables $\theta_{i}$ with values taken uniformly at random from *A*. *l* random variables $x_{i}$, 2N+4cN random variables $r_{i}$, and $\left|D\right|$ random variables $d_{i}$ with values taken uniformly at random from $\left\{0,1\right\}$. A binary string s of length at most 2N+4cN for recording true measurement outcomes related to Bob’s measurement outcomes, where s is initially set to be vector 0.(5) A fixed function $C\left(i,\phi_{i},\theta_{i},r_{i},x_{i},s\right)$ that for each non-output qubit *i* computes the angle of the measurement of qubit *i* to be sent to the Bob.**Initial step:**(1) Alice’s move: Alice sets all the values in s to be 0 and encodes the *l*-qubit input state as $|e\rangle =X^{x_{1}}Z\left(\theta_{1}\right)⊗\cdots ⊗X^{x_{l}}Z\left(\theta_{l}\right)|I\rangle$. She then prepares the remaining qubits in the following form: If i∈D, then qubit *i* is set to be $|d_{i}\rangle$; otherwise qubit *i* is set to be $\prod_{j\in N_{S\left(G\right)}\left(i\right)\cap D}Z^{d_{j}}|{+}_{\theta_{i}}\rangle =|{+}_{\theta_{i}+\sum_{j\in N_{S\left(G\right)}\left(i\right)\cap D}d_{j}\pi }\rangle$, where $N_{S\left(G\right)}\left(i\right)$ represents the neighborhood of vertex *i* in S(G). Then Alice sends Bob all qubits in the order of the labeling of the vertices of the graph S(G).(2) Bob’s move: Bob receives 2N+4cN single qubits and entangles them according to S(G).**Step *i*: $1\leq i\leq \left(2N+4cN\right)$**(1) Alice’s move: Alice computes the angle $\delta_{i}$ equal to $C\left(i,\phi_{i},\theta_{i},r_{i},x_{i},s\right)$ and sends it to Bob. If qubit *i* is the trap qubit, then the angle $\delta_{i}$ is set to be $\theta_{i}+r_{i}\pi$.(2) Bob’s move: Bob measures qubit *i* with angle $\delta_{i}$ and sends Alice result $b_{i}$.(3) Alice’s move: Alice sets the value of $s_{i}$ in s to be $b_{i}⊕r_{i}$.**Verification:**(1) After obtaining all the output qubits from Bob, if the trap qubit *t* is an output qubit, Alice measures it with angle $\delta_{t}=\theta_{t}+r_{t}\pi$ to obtain $b_{t}$.(2) Alice accepts if $b_{i}=r_{i}$ for all the trap qubits *i*.(3) Alice applies corrections according to measurement outcomes $b_{i}$ and secret parameters $\theta_{i}$, $r_{i}$ at the output layer green qubits in order to obtain the final output.

**Theorem** **1**(Correctness). *If Alice and Bob follow the steps of Protocol 1 honestly, then Alice accepts the correct outcome.*

**Proof.** Proof follows along similar lines of Theorem 2 in [25]. In Protocol 1 the dummy qubits are placed at white vertices of the coloured sandglass-like graph S(G). Note that the effect of dummy qubits is the break operation on the graph S(G). As a result, a green computation subgraph and a red trap subgraph are obtained. Since both subgraphs have no effect to each other, we consider the measurements on the computation subgraph and the trap subgraph separately.The correctness on the computation subgraph stems from the correctness of universal blind quantum computation [2]. To clarify, if each qubit in computation subgraph is rotated qubit $|{+}_{\theta_{i}}\rangle$ and measured in angle $\delta_{i}=C\left(i,\phi_{i},\theta_{i},r_{i},x_{i},s\right)$, then all the deviations from the actual implementation of the measurement pattern are corrected. Therefore Alice will get desired computation output.As for the trap subgraph, trap qubits are isolated. Every trap qubit $|{+}_{\theta_{i}}\rangle$ will obtain deterministic measurement outcome $b_{i}=r_{i}$ after it is measured in angle $\delta_{i}$ equal to $\theta_{i}+r_{i}\pi$. Alice will accepts the output, as honest Bob will always return $b_{i}=r_{i}$ for all trap qubits. □

**Theorem** **2**(Verifiability). *Protocol 1 is 0.905 verifiable in the case of quantum output and 0.866 verifiable in the case of classical output.*

The proof of above Theorem 2 can be found in Appendix A. According to the process of proof, the verifiability of Protocol 1 is equivalent to solving the following optimization problems.
(1)$$\begin{array}{cc} & \min_{\alpha \in (0,1)}\max\left\{1-\frac{1}{4}\alpha ,1-\frac{1}{2}\alpha ,-\frac{1}{4}\alpha^{2}+\frac{1}{2}\alpha +\frac{3}{4}\right\}, \end{array}$$
(2)$$\begin{array}{cc} & \min_{\alpha \in (0,1)}\max\left\{1-\frac{1}{2}\alpha ,-\frac{1}{4}\alpha^{2}+\frac{1}{2}\alpha +\frac{3}{4}\right\}, \end{array}$$
where Equations (Equation 1) and (Equation 2) respectively correspond to the case of quantum output and the case of classical output. Theorem 2 shows that the probability of accepting an incorrect outcome is constant.

## 5. Optimization of Verifiability

While we achieve the verification of blind quantum computing with the sandglass-like resource state, our protocol’s verifiability is too high to be applied in practice. In this section, similar to [25], we will utilize one method respectively to reduce verifiability ϵ to arbitrarily small number in both cases of classical output and quantum output.

In the case of classical output we repeat Protocol 1 a certain number of times. Since all repetitions obtain the same correct output when Bob is honest, the verifiability can be decreased by adding an additional verification condition that Alice accepts final output if all of repetitions get the same output. From this result we can construct a new verification protocol based on repetitions, as given in Protocol 2. In the case of quantum output we use the technology of fault-tolerant code [33,34], which is often used in topological fault-tolerant blind quantum computation [4]. The main idea is that malicious Bob needs to make more attacks on computation qubits because of the existence of fault-tolerant code, which will increase the possibility of being caught by Alice. We therefore have Protocol 3.
**Protocol 2** Optimised VUBQC with sandglass-like resource state for classical output.**Alice’s resources:**(1) The number of repetitions $R=\frac{\log\epsilon }{\log0.866}$, where ϵ is the desired security level.(2) The rest of the resources are the same as Protocol 1.**Step *i*: 1≤i≤R**(1) Follow the steps of Protocol 1, where each repetition of Protocol 1 corresponds to identical computation task.(2) If Alice accepts the output, she records the classical output as $O_{i}$.**Verification:**(1) If any single repetition of Protocol 1 is rejected, the overall computation will be rejected. Otherwise, Alice compares all $O_{i}$. If all $O_{i}$ are identical, Alice accepts this output as the output of computation.

**Theorem** **3.**
*Protocol 2 is $0.866^{R}$ verifiable for classical output, where R is the number of repetitions.*


**Proof.** Recall that the verifiability represents maximal probability that Alice accepts an incorrect outcome. Because the condition that Alice accepts final output is that all repetitions of Protocol 1 are accepted and all of them return the same output. The event that Alice accepts an incorrect output means that all repetitions of Protocol 1 are accepted and return the same incorrect output. Since the verifiability of Protocol 1 is 0.866, the verifiability of Protocol 2 is $0.866^{R}$. □

**Protocol 3** Optimised VUBQC with sandglass-like resource state for quantum output.

**Alice’s resources:**

(1) A base graph *G* encoded in a fault-tolerant way for correcting errors less than δ.
(2) The rest of the resources are the same as Protocol 1.

**Same steps as in Protocol 1.**



**Theorem** **4.**
*Protocol 3 is $0.905^{\left\lceil \frac{\delta }{2\left(2c+1\right)}\right\rceil }$ verifiable for quantum output, where δ is the number of tolerated errors on the base graph G and c is the maximal degree of G.*


The proof of the above Theorem 4 can be found in Appendix B. From Theorem 3 and Theorem 4 we can see that the verifiability of verification protocol is exponentially small.

## 6. Conclusions

Inspired by the dotted triple-graph by Kashefi and Wallden [25], we have introduced the concept of sandglass-like graph whose corresponding graph state can be used to be the resource state of verifiable blind quantum computing. We then proposed a verifiable blind quantum computation protocol with sandglass-like resource state. Based on this protocol, we proposed one new scheme in the case of classical and the case of quantum output to improve the verifiability of the original protocol.

Our main contribution can be described as follows. We have broken the symmetry of the trap-coloring in [25]. In other words, the possibility to be colored in green, the possibility to be colored in white, and the possibility to be colored in red are set to be not equal for each primary vertex. This essential point allows us to design a better resource state, which only requires a less linear overhead in the size of the computation. In addition, we achieves a better verifiability for the case of classical output, i.e., a lower probability that the client accepts a wrong outcome from the server, by optimizing the setting of the probability in the trap-coloring.

In [17], Joseph F. Fitzsimons et al proposed a VUBQC protocol using a dotted-complete graph state. Their verifiability is ${\left(5/6\right)}^{\left\lceil 2d/5\right\rceil }$, where *d* is the defect thickness under RHG fault-tolerance scheme [35,36,37]. Here, RHG fault-tolerance scheme is a fault-tolerant version of the one-way quantum computer using a cluster state in three spatial dimensions, which was proposed by Raussendorf, Harrington and Goyal [36]. However, the overhead of their protocol is quadratic. In other words, the number of qubits required for the protocol is $O(N^{2})$, where *N* is the number of qubits used to implement desired computation. The protocol of Elham Kashefi et al [25] considered a dotted triple-graph state. Their verifiability is ${\left(8/9\right)}^{R}$ in the case of classical output and ${\left(\frac{8}{9}\right)}^{\left\lceil \frac{\delta }{2\left(2c+1\right)}\right\rceil }$ in the case of quantum output, where *R* denotes the number of repetitions, δ is the number of errors that can be detected or corrected, and *c* is the maximal degree of the base graph *G* implementing desired computation. In addition, their overall cost is 3N+9cN. In contrast to these schemes, our protocols’ verifiability is $0.866^{R}$ in the case of classical output and $0.905^{\left\lceil \frac{\delta }{2\left(2c+1\right)}\right\rceil }$ in the case of quantum output. It means that our verifiability is better in the former case and slightly worse in the later case. More importantly, our overhead is 2N+4cN.

For future studies, our construction can be applied to device-independent VUBQC [20,21,24] and other specific fault-tolerance codes. It is still an open problem to further reduce overhead of VUBQC.

## Figures and Tables

**Figure 1 entropy-22-00996-f001:**
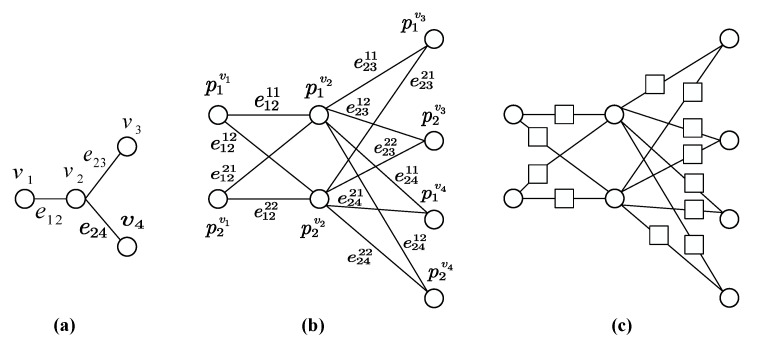
(**a**) A base graph *G* consisting of four vertices and three edges. (**b**) The intermediate graph I(G) corresponding to the base graph G. (**c**) The sandglass-like graph S(G) corresponding to the base graph G. The circle represents a primary vertex and the square represents an added vertex.

**Figure 2 entropy-22-00996-f002:**
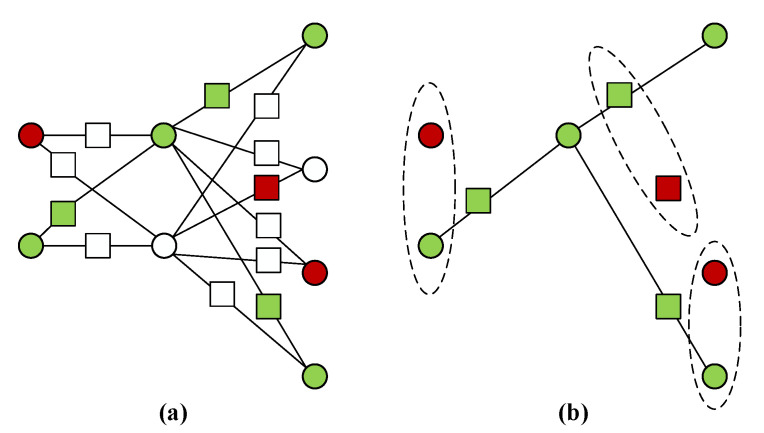
(**a**) The trap-colouring of the sandglass-like graph S(G). (**b**) A computation subgraph and a trap subgraph obtained by performing break operations on the white vertices of the coloured S(G). For each green computation vertex, there may be a corresponding red trap vertex.

**Figure 3 entropy-22-00996-f003:**
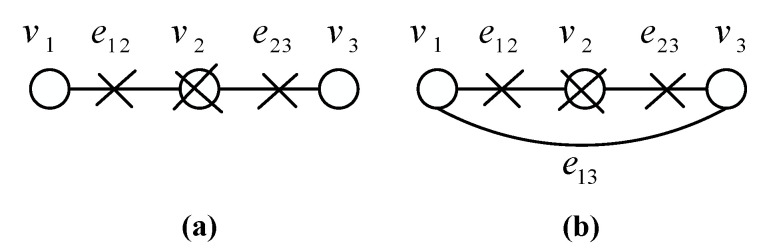
(**a**) A break operator on the vertex $v_{2}$. (**b**) A bridge operator on the vertex $v_{2}$.

**Figure 4 entropy-22-00996-f004:**
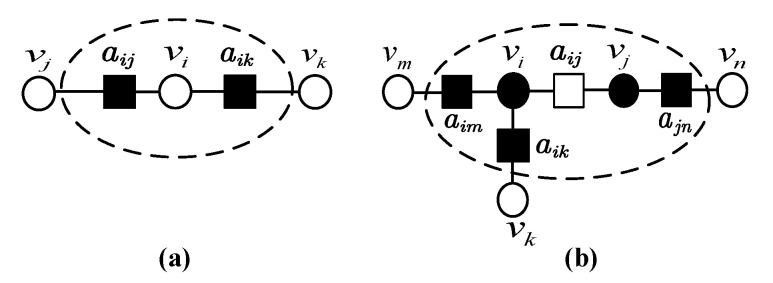
(**a**) The introduction of a primary base-location $v_{i}$. (**b**) The introduction of an added base-location $a_{ij}$. Black vertices are excluded for satisfying independently colourable locations (ICL). The dash line circles all the influenced vertices.

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
