# Peer review of "Improved Resource State for Verifiable Blind Quantum Computation"

_entropy, 2020, doi:10.3390/e22090996_

Round 1

Reviewer 1 Report

This manuscripts presents a protocol for the verification of blind quantum computing using sandglass-like graph state as the resource state. Trap qubits are incorporated into graph state along with computation qubits to verify the outcome of the computation while remaining private to the quantum computing server. The manuscript highly relies on Refs. 17 and 25. Despite comparable verifiability with those, it claims less overhead. The presented work including protocols and proofs sound reasonable. However, I have the following comments:

1- Although it seems that the term "verifiable" is used in previous work as well, this sounds more like "authenticating" the server whether or not it's quantum. Verification should be checking whether the output from quantum computer is correct or not. Even if Alice may not detect any issue based on her trap qubits, the outcome may be still incorrect. Because it may require verification by other means especially if the computational task is intractable classically or the problem is not a trapdoor problem (e.g., factorization).

2- Please proof read the manuscript carefully, as there are many errors.

3- Page 3: please clearly distinguish D(G) and S(G).

4- When referring to Protocol 1, etc., please mention where they're actually given. Nowhere in the manuscript says it's given at the bottom of the manuscript. This was confusing initially as it was not clear from main text where these protocols were.

5- Page 7: define RHG.

Author Response

Thank you for your valuable comments, positive reviewing, and reference recommendations.

  1. The term "verifiable" in the context of VUBQC originates from the notions of completeness and soundness in the context of interactive-proof system. Given a promblem that is classically intractable, the verifier can accept a correct solution with high probability and reject a invalid solution with high probability at the end of interaction with the prover. Note that even if the verifier accept the outcome sent by the prover, the outcome may be still incorrect. However, the probability that the verifier accepting a wrong outcome can be reduced to a value approaching 0 through some improvements for VUBQC. A supplementary explanation has also been given in the section 1.
  2. We have corrected some grammatical and symbolic errors.
  3. We have defined an intermediate graph I(G) in the construction procedure of the sandglass-like graph S(G). Since S(G) is obtained by performing the dotting operator on I(G), the sandglass-like graph S(G) is actually equal to D(I(G)). We have updated Figure1 and Figure 2 for a better understanding of the construction and the coloring.
  4. We have moved protocol 1, etc. to the corresponding positions of the main text.Protocol 1 can be found in the line 235. Protocol 2 can be found in the line 306. Protocol 3 can be found in the line 326.
  5. In section 6 we have briefly introduced the RHG fault-tolerance scheme proposed by Raussendorf, Harrington and Goyal.

Reviewer 2 Report

I reccomend to accept the paper in present form

Author Response

We thank for your recommendation of publication in Entropy.

Reviewer 3 Report

The authors deal with the problem of quantum verification schemes in the context of blind quantum computing. They present a resource state that uses less qubits than previous approaches.

The procedure to constuct the sanglass graph is not so clear in this manuscript. I had to see what is done in Ref. 25, in order to follow what is done here (a similar remark holds in many parts of the manuscript, due to the bad quality of the presentation). The example is clear in Fig. 1, but the construction for a base graph of size N is not clear. Perhaps, the authors could try to improve the notation (or the description). Furthermore, it seems to me that the notation in the description of the sandglass graph does not coincides with the one given in Fig. (1) (please, check this).

In page 6, the authors refer to “Protocol 1”. But it is not clear where to find it. Of course, it is included after the references. My suggestion is that the refer to it as “Protocol 1 shown In Appendix 1”, or something like that. If an Appendix cannot be added, I suggest to include the protocols in the corpus of the text, in order to smplify the reading of the manuscript.

In general, I find two main problems in this manuscript:

  • The first one is that it is not suitably presented. The English style is sometimes difficult to follow, and the notation is not clear in many points.
  • The results presented here are very similar to those of previous papers (specially, to those of references 17 and 25). It is hard to tell the difference. Perhaps I missed something, but it would be better if the authors clarify what is really new about their work. In other words, I don’t see why this is substantial contribution, given that it is a small variant of previous approaches. If the contribution is just to reduce the number of qubits, perhaps, the presentation should be more focused on that point. It was more clear for me to read the previous papers (17 and 25), tan the information provided by the authors of this manuscript.

I suggest that the authors try to improve the presentation of the manuscript, and that they try to explain more clearly why this should be considered a relevant scientific result (given that contribution is very similar to that of previous approaches).

Author Response

Thank you for your valuable comments, positive reviewing, and reference recommendations.

  1. We have updated the construction of the sandglass-like graph S(G) corresponding to an arbitrary base graph G, where the definition of an intermediate graph I(G) has been supplied. We have adapted Figure 1 and Figure 2 for a better understanding of the construction and the coloring, where Figure1 gives an example of the construction for a base graph G consisting of four vertices and three edges.
  2. We have moved protocol 1, etc. to the appropriate positions of the main text. Protocol 1 can be found in the line 235. Protocol 2 can be found in the line 306. Protocol 3 can be found in the line 326.
  3. We have corrected some grammatical and symbolic errors. We have additionally explained the notions of the term "verifiable" in section 1. The definition of the graph state |G⟩ corresponding to a graph G has been supplied in section 2. We have supplied the definition of primary vertices and the definition of added vertices in the line 127.
  4. Our main contribution can be described as follows. We have broken the symmetry of the trap-coloring in [25]. In other words, the possibility to be colored in green, the possibility to be colored in white, and the possibility to be colored in red are set to be not equal for each primary vertex. This essential point allows us to design a better resource state, which only requires a less linear overhead in the size of the computation. In addition, we achieves a better verifiability for the case of classical output, i.e. a lower probability that the client accepts a wrong outcome from the server, by optimizing the setting of the probability in the trap-coloring. A supplementary explanation has also been given in the section 6. In this paper, we have overcome the open problem that exploring a verification protocol with a smaller verifiability while keeping the cost of resource optimal.

Round 2

Reviewer 3 Report

The authors have prepared a revised version where they explain in a better way what they have achieved here. Furthermore, they have improved the presentation in some crucial parts of the manuscript.

I suggest publication of the manuscript. But the proofreading should be very careful, as many problems in the presentation are still present (typos, English style problems, etc).